# Green Infrastructure Planning in Metropolitan Regions to Improve the Connectivity of Agricultural Landscapes and Food Security

**Carolina Yacamán Ochoa *** , **Daniel Ferrer Jiménez and Rafael Mata Olmo**

Research Group Landscape and Territory in Spain, Mediterranean Europe and Latin America, Department of Geography, Autonomous University of Madrid, 28049 Madrid, Spain; daniel.ferrer@uam.es (D.F.J.); rafael.mata@uam.es (R.M.O.)
* Correspondence: carolina.yacaman@uam.es

**Abstract:** Green infrastructure (GI), as a concept and as a tool for environmental land-use planning at various scales, has burst onto the academic, political, and policy-making scenes in the last two decades. This tool, associated with strategic planning, offers integrated solutions for improving the ecological connectivity and urban resilience of open spaces, especially those affected by processes of urban sprawl, the abandonment of agriculture, and the territorial fragmentation of habitats and traditional agricultural landscapes. In spite of the advantages of GI, its design and implementation face a range of challenges and limitations. In this context, this paper has two objectives: Firstly, to address a critical review of recent literature on the subject, which, among other things, highlights the lack of references to the role of peri-urban agriculture in GI planning, and the positive contribution made by peri-urban agriculture to the local food supply and other regulatory and cultural services. Secondly, to propose a methodology to contribute to integrating practical GI planning in metropolitan regions to maximize the activation of traditional agricultural landscapes and the improvement of landscape connectivity in metropolitan regions for the reconnection of rural-urban relationships.

**Keywords:** landscape ecology; metropolitan planning; multi-functionality; peri-urban agriculture; food security; urban resilience

## 1. Introduction

Green infrastructure (GI), as a concept and as a tool for environmental land-use planning at various scales, has burst onto the academic, political, and policy-making scenes in the last two decades. GI has proved to be useful for the planning of open spaces in metropolitan areas and urban regions because of its integrative perspective and multi-functional focus [1–6]. Open space is a term which can be used in a broad sense to describe areas with a low level of human intervention. These undeveloped areas of land are increasingly becoming key pieces of metropolitan planning [2]. Regarding open space planning, the provision of ecosystem services stands out in GI development [7–12]. This approach strengthens urban resilience [13–15] and facilitates the transition toward sustainable land use at a regional scale [16,17].

However, it is necessary to draw attention to the fact that a conceptual and methodological gap exists in the literature on this subject. This gap concerns the important role played by agriculture, especially in peri-urban areas, in delivering a wide variety of ecosystem services, especially in the face of climate change and intensive land-use changes. Only Feria and Santiago [18]; Roc et al., 2020 [19]; Schmidt and Hauck [20]; and Yacamán and Mata [21] explicitly mention the need to improve the protection and management of agricultural landscapes in GI. This is despite the fact that sustainable land planning can enhance all the ecosystem services delivered by metropolitan green infrastructure,

especially food security, as well as improving agricultural productivity and profitability and urban resilience [19]. Meerow and Newell [22] (p. 45) proposed the following definition of urban resilience that is appropriate for the context of this article: "Urban resilience refers to the ability of an urban system—and all its constituent socio-ecological and socio-technical networks across temporal and spatial scales—to maintain or rapidly return to desired functions in the face of a disturbance, to adapt to change, and to quickly transform systems that limit current or future adaptive capacity". Accordingly, in order for GI to be able to simultaneously cover the provision of a wide spectrum of ecosystem services and strengthen urban resilience, Feria and Santiago [18] emphasize the fact that it is necessary to reverse the secondary role assigned to agriculture in both the application of the concept of GI and in urban planning practice in general. From this perspective, GI must strengthen sustainable agricultural land use from a multi-functional perspective through the creation of specific guidelines for the activation of peri-urban agriculture. This ensures the production of fresh, healthy, local products and the preservation of land with a high agroecological value, while ensuring that the local population can access the landscape and enjoy it [21].

Nevertheless, in the current context of globalization and the liberalization of land policies, urban sprawl continues to be one of the major challenges faced by regional planning in the twenty first century [23]. Its negative effects, especially clear in the dynamics of land-use change [24], are caused by multi-dimensional drivers that cause multi-dimensional economic, social, and ecological impacts [1]. In this process of unsustainable urban expansion, the demand for urban growth and the increase in road infrastructure tend to predominate [25]. This reduces the availability of open spaces and access to them, which are essential components for maintaining the health and wellbeing of urban residents as these spaces provide multiple environmental and social benefits [6]. This problem intensifies with the speculative processes and the expectation of huge economic gains generated by the reclassification of rural land to urban land in the areas close to cities, with the ensuing loss of fertile soil and the collapse of the market for agricultural production land [26]. Therefore, a major part of the strategic dimension of GI must be focused, on one hand, on the promotion of compact urban development using innovative planning, and, on the other hand, on the improvement of environmental quality and socio-economic and ecological processes by involving policy makers and through stakeholder engagement [27]. In order to achieve this, it is necessary to focus on developing policies which promote the transition to more efficient forms of land management, giving greater weight and significance to open spaces through protection and rezoning and rural–urban reconnection [28], and with the activation of its multiple ecosystem services, especially the provision of fresh food.

As an alternative, Borelli [29] and others [7,13] propose integrating a landscape approach into spatial planning activities to properly face complex and widespread environmental, social, and political challenges that transcend the limits of traditional management. This requires, firstly, zoning at a landscape scale, which allows the development of knowledge on the biophysical and cultural processes of non-developable land, on endogenous resources, material and symbolic services and values. Subsequently, it is necessary to organize and manage, depending on their functional capabilities, the different landscape units. These landscape units are part of the territorial matrix and their organization and management aim to favor a mosaic of different opportunities, prioritizing those which are perceived by local communities to be part of their cultural identity heritage [21].

The current state of GI planning in Europe, which appears in the peer-reviewed literature, shows that GI has been incorporated into land-planning procedures at different scales, ranging from local to regional and supra-municipal scales [30–32]. Although there is no internationally accepted position on the most appropriate scale for planning GI, a number of authors have suggested that the regional and sub-regional scales are the best fit for the sustainable management of open spaces subjected to the impacts of urban sprawl [33–35], and, as in our case study, affected by processes of the abandonment of agriculture and the increasing fragmentation of agricultural areas. Feria and Santiago [18] indicate that as the dynamics and pressures derived from the processes of urban sprawl

go beyond the municipal scale, it is necessary to re-dimension this phenomenon at a supra-municipal scale to implement the proper planning of open spaces articulated by GI.

In accordance with all these aspects, the proactive incorporation of agricultural space into GI is essential, especially in Mediterranean cities, given their strong traditional ties to their agricultural surroundings [36]. Therefore, from the perspective of renewed agri-food policies and the commitments made by the European Landscape Convention of the Council of Europe [37], the conservation and dynamization of agricultural landscapes from a multi-functional and territorial perspective is both an opportunity and a challenge to enhance the different ecosystem services which the traditional landscapes provide.

Based on the lack of attention paid, in both academic research and public policies, to the role of peri-urban agricultural landscapes in GI, this research presents a methodology for the planning of GI, which enhances landscape connectivity and maximizes agricultural landscapes benefits. This aims to ensure the permanence and regional connectivity of open spaces and conserve landscape diversity in metropolitan areas. The methodology provides an opportunity to increase the importance of peri-urban agricultural spaces due to their multi-functional dimension. It highlights their strategic role in the production of quality local food, and in the construction and management of landscapes that attribute territorial identity to food and promote the reconnection of rural–urban relationships.

## 2. Materials and Methods

### 2.1. Study Area

This research is centered on the "Comarca de las Vegas". It is a case study that will be used to develop and evaluate the problems and potential of the methodology. The study area is located in the south east of the Autonomous Region of Madrid (Figure 1). There were two main reasons for the selection of this functional unit: firstly, it is the agricultural region with the largest area of production in the Autonomous Region of Madrid; and, secondly, the need to develop territorial planning at a supra-municipal scale that prevents the increase in land use by urban development and land fragmentation, which inevitably causes the degradation of ecosystems and the loss of fertile soil.

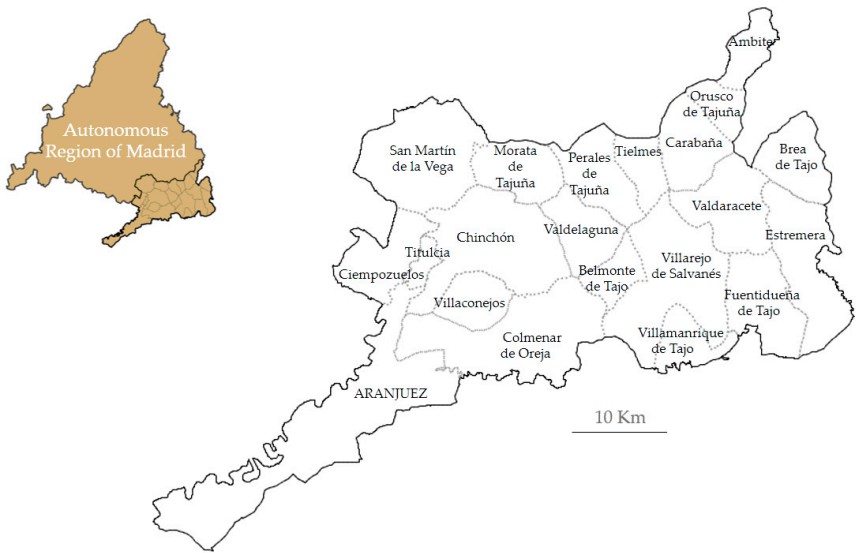

**Figure 1.** The study area.

This extensive territorial unit comprises 21 municipalities, covering 1308.03 ha (approximately 16% of the regional total). According to data from 2019 (INE), 153,632 inhabitants live in the area (2.3% of the regional total), and they are mainly concentrated in the municipalities of Aranjuez

(59,607 inhabitants), Ciempozuelos (24,592 inhabitants) and San Martín de la Vega (19,170 inhabitants). The average density is 117 inhabitants/km$^2$, though this figure varies greatly between the municipalities. Las Vegas has a rich farming and agricultural history. Its main agroecosystems are irrigated arable crops located on the *vegas* (Calcareous plateau with herbaceous crops) of the River Tajo and its tributaries Jarama y Tajuňa and the mosaic of rain-fed arable crops, olive groves, and vineyards, which are distributed over the *páramos* (Calcareous plateau with herbaceous crops), *campiñas* (Sedimentary hills with herbaceous crops and open fields), and *vertientes* (Sedimentary slopes).

The relief, resulting from vigorous erosion processes, is intrinsically related to the shaping action of the previously mentioned river courses. It is characterized by the presence of three large geomorphological units, namely: the *vegas*, the *páramos*, and the *vertientes*. In general, the Comarca de las Vegas is characterized by a topography which is not particularly rugged in most of the region (*vegas* and *páramos*), with an altitudinal range between 400 and 850 m and a gradient which increases on the slopes that connect the other two domains.

Given the long agricultural tradition of this region, the forest ecosystems do not cover a large surface area, even though there is a striking mosaic of vegetation formations that are very valuable in ecological and landscape terms, of which some well-conserved shrubs of once present extensive *sotos de ribera* (Riverside groves) and *dehesas* (Pastureland Mediterranean open oak woodland with pastures) remain. There are also mountains covered in oaks, different types of *matorral* (Scrubland); spots of different types of conifers and, on a different scale, vegetation belonging to the *aljezares* (Gypsiferous escarpment) which hold great botanical interest [38]. The region has a large extension of areas of protected rural and natural terrain.

The study of this region in relation to its strategic territorial planning is of great interest, given that this region has historically played an important role as an area of food supply for the metropolitan region. This role has been boosted by the availability of water and the presence of *vegas* with a high agricultural capacity. These circumstances encourage the development of horticultural agriculture of exceptional quality, making it the strategic "pantry" of Madrid [38]. Nevertheless, the lack of committed territorial planning that considers these values, and the heterogeneity and obsolescence of municipal urban planning [26] have transformed the region into a space in which agricultural spaces exist alongside the new features (theme parks, health and leisure facilities, rural houses for tourism, hospitality establishments, etc.) and road infrastructures which the metropolitan area demands, without any clear territorial and landscape criteria.

Since the mid-twentieth century, agricultural production has varied substantially, especially due to the loss of profitability of small-scale family farms. This is a consequence of the fact that the markets of Madrid are supplied with products from large scale intensive farms with more competitive prices, as their crops are cultivated under plastic in the south of Spain [39]. Metropolitan dynamics are severely altering the territorial matrix and limiting the food production capacity. These dynamics include second homes and hobby farming [40], as well as the pressures coming from some illegal construction on rural land and, fundamentally, mining, roads, and land use for the construction of leisure and industrial facilities. In just ten years (1999−2019), the percentage of individuals contributing to the social security system in the field of agriculture in las Vegas has decreased by 20.85% (INE), and this is considering that the numbers were already low in 1999. This indicates that there is a clear risk that the traditional agricultural sector will collapse and disappear unless the whole agricultural cycle of the urban region of Madrid is reactivated.

Nevertheless, in spite of the changes identified in land use and types of crops, las Vegas continues to be the region with the largest surface area of crops in the Autonomous Region of Madrid. The large amount of fertile land is still well conserved despite the notable regression in horticultural activity. As a whole, las Vegas still maintains an extraordinarily high level of landscape interest and a high heritage value. Among the places which stand out in this region, are in the municipality of Aranjuez, some areas of which, next to the royal palace and its gardens, were declared as Cultural Landscapes on the UNESCO World Heritage List. These natural and agricultural values make planning and

management focused on a systemic approach necessary for the recuperation of some distinguishing historical rural landmarks and their horticultural production [41]. It is also necessary to stop the urban pressure that can lead to a situation that can seriously compromise the food production capacity.

*2.2. Conceptual Framework*

The methodology we propose is based on the key ideas we observed from the systemic literature review [42] that the authors previously carried out in the framework of the research project "*Paisaje y Huerta de Madrid*" (Landscape and *Huerta* of Madrid): a green infrastructure that contributes to the protection of agricultural spaces in the south-east of the Autonomous Region of Madrid (Comunidad de Madrid). This critical review clearly shows that most of the recent research on GI (2015–2020) mainly focuses on two issues: ecosystem services and the treatment of ecological connectivity in open spaces. We identified a research gap related to the role that agrarian multifunctional landscapes can play in boosting the resilience of the socio-ecological matrix and developing sustainable regional food systems. These landscapes must be considered when planning open spaces for the maintenance of the provision of a wide spectrum of ecosystem services and landscape connectivity. In this sense, it is necessary to go into greater depth on the issues that are highlighted below to improve the protection and ecological and territorial connectivity of multi-functional landscapes inserted into the territorial matrix of the GI:

- Promote multi-scale planning to prevent municipal administrative boundaries from becoming a barrier to the biophysical functioning of ecosystems [33–35].
- Use a multi-stakeholder approach to improve governance and active participation with the objective of avoiding a disconnection from the interests of the local population and to cope with the specific demands of the regions [43,44].
- GI implementation in agricultural landscapes requires reliable and flexible measures that suit farming practices and that are effectively communicated to farmers, ensuring better coordination and cooperation at a landscape level to enhance their effectiveness [20] (p.899).
- Integrate an embedded multi-functional approach and the four dimensions of resilience-policy, performance, connectivity, and social issues, in all of their forms—institutional, climatic, economic, and ecological—when planning the GI network [10,13–15].
- Reinforce the role of traditional agricultural landscapes that make up the territorial matrix into which the GI is integrated [18–21]. This is especially relevant in peri-urban areas under pressure from urban sprawl. Therefore, it is important to define an integrative multi-scale methodology to promote economically viable and environmentally sustainable agriculture.
- Understand the territorial matrix as a socio-ecological system to improve the management of social and ecological processes and relationships. This ensures ecosystem services, in particular the landscape services, of the entire system [45,46].
- Promote new government frameworks that go beyond biodiversity protection in official natural parks and nature preserves, by promoting truly holistic GI development that is able to conserve the natural, cultural and landscape heritage as a whole.

*2.3. The Methodology Proposed for Mapping the GI*

The methodology proposed is based on landscape ecology and its ability to adequately address the complex relationships between humans and their environment. It also uses a multi-scalar approach for the development of multi-functional GI to foster ecological, productive, and cultural functions. The design and analysis are performed using geographical information systems (GIS), together with participatory methods and expert assessments. The three major challenges which are faced, and will hopefully be solved through GI planning from a multi-functional perspective on a sub-regional (or supra-municipal) scale, are (i) improvement of the connectivity of the habitats which are outside the protected natural areas in order to strengthen the ecosystem services they offer; (ii) improvement

of the quality of agricultural landscapes that make up the territorial matrix; (iii) prevention of urban pressures which fragment the landscape; and, finally, (iv) promotion of multi-functional landscapes and their contribution to the strengthening of the local-regional food system. These issues are critical for ensuring the functionality of the GI and the improvement of the provision of ecosystem services, especially food supply, thus ensuring urban resilience and maintaining the character and coherence of traditional landscapes.

2.3.1. Mapping the GI Network

Our methodological approach consists of seven steps (Figure 2):

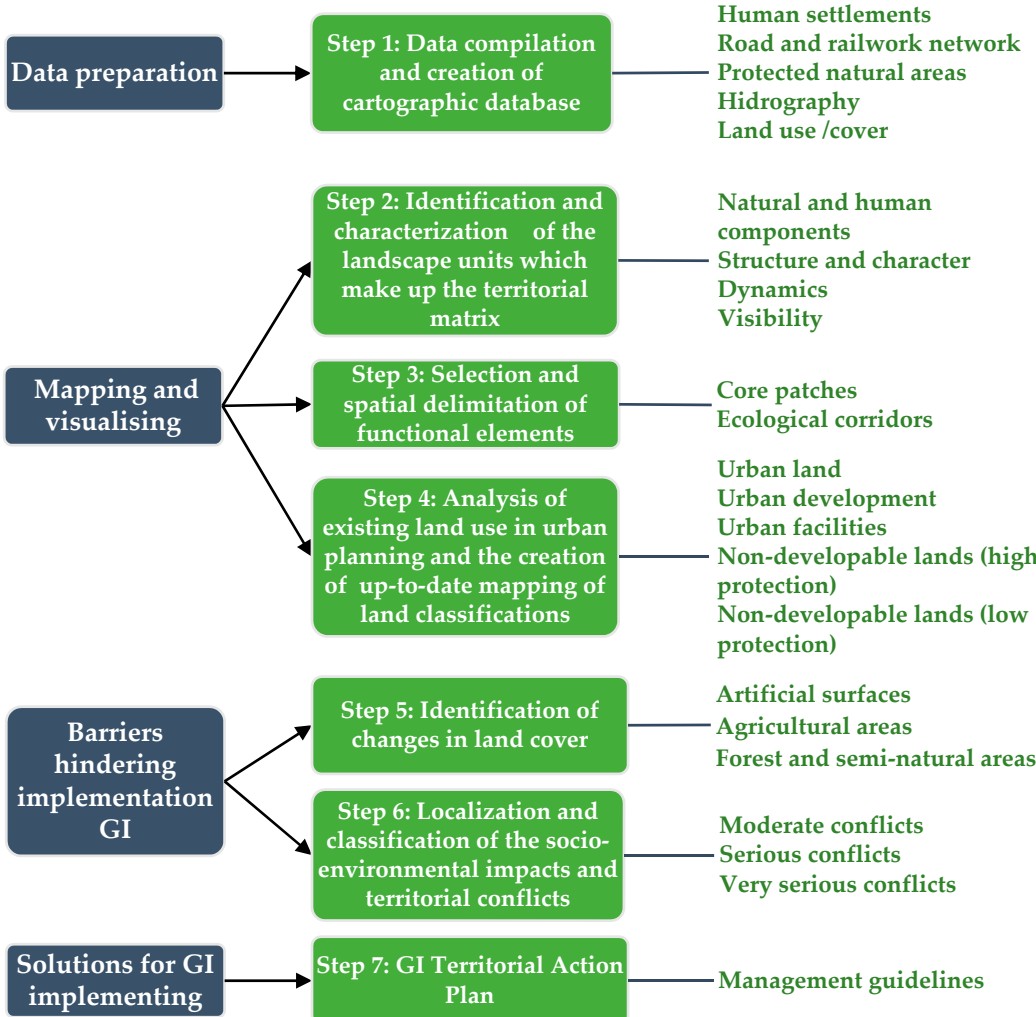

**Figure 2.** Methodology for Green Infrastructure planning based on landscape ecology approach.

Step 1: Data compilation and creation of a database that is sufficiently detailed to be used at a supra-municipal scale.

All data compiled needs to be suitable (scale and resolution) for the entire GI mapping and the information must be useful for decision making. The working scale used for all the mapping work indicated was 1:25:000. We compiled data on topographic information, human settlements, road, and railway networks, protected natural areas, hydrography and land use and land cover.

Step 2: Identification and characterization of the landscape units which make up the territorial matrix of the selected study area.



The objective is to carry out a characterization, diagnosis, and evaluation of the quality of the landscape of the study area that allows the establishment of criteria for future protection and spatial planning. The method used for the identification and characterization of the landscape units corresponds to the following criteria [47]:

- Natural and human components which make up the landscape: the physical and human features and elements which are best reflected in the morphology, functioning, and dynamics of the landscape are listed and described.
- The structure and character of the landscape: the different elements of the configuration of the unit are discursively articulated and integrated, leading to a reading of the character of the unit, the singularity of which stands out when compared to others.
- The dynamics of the landscape: the processes and active dynamics in each unit are identified and characterized using a historical analysis of the evolution of the landscape and its most recent transformations.
- The visibility of the landscape: the characteristics of the landscape are analyzed from a visual perspective, integrating the views on different levels which can be obtained from the landscape unit, incorporating the most significant visual features into the characteristics of each unit.

Step 3: Selection and spatial delimitation of the functional elements for the formalization of the GI.

The objective is to define core patches and ecological corridors to strengthen and restore landscape connectivity. The identification of core patches and corridors must be based on the functions that they have in the network (See Section 2.3.2). Information from different organisms and public entities should be collated and analyzed. Information can be downloaded in a vectorial format (shp and the GIS geodatabase) and in a raster format. This information can be complemented with fieldwork and the consultation of secondary sources. For the improvement of the spatial and territorial connectivity, the selection of the core patches can be based on the landscape ecological planning approach where other areas apart from the protected natural spaces should be considered, in view of the fact that protected areas are neither numerous enough nor large enough to maintain and improve spatial, territorial and ecological connectivity [48]. For the identification of the ecological corridors, the study of the permeability for the movement of target species has been proposed [49]. This methodology consists of modeling an itinerary that requires the smallest amount of movement for a group of target species and habitats studied.

Step 4: Analysis of existing land use in urban planning and the creation of up-to-date mapping of land classifications.

The objective is to determine the real capability of the region to adopt, in urban planning terms (land classification and rating) the GI proposal. This analysis will detect any possible conflict points and areas. The analysis of urban planning is a priority as it is the main planning instrument for establishing land use at a local scale.

Step 5: Identification of changes in land cover in the study area.

The objective of this step is to identify the dynamics of land cover, fragmentation and loss of habitats, and the abandonment of agricultural activity. This information can be obtained by using the spatial datasets from the European CORINE Land Cover project for different years. Given the wide variety of sub-categories available on land cover in CORINE, we gather data on changes in three land cover categories of the extent of cropland, forestry, and artificial land, as well as information on land-use changes to urban and artificial land (Table 1). Since agricultural abandonment is poorly represented in the land cover data captured by CORINE [50,51], we contrast the information with fieldwork and with agricultural statistics.

**Table 1.** Levels of land-cover between 1990 and 2018.

| Land Cover Categories | Description |
|---|---|
| Artificial surfaces | Urban fabric; industrial, commercial and transport units; mines, dump and construction sites; artificial, non-agricultural vegetated areas |
| Agricultural areas | Arable land; permanent crops; pastures; heterogeneous agricultural areas |
| Forest and semi-natural areas | Forest; shrub and/or herbaceous vegetation associations; open spaces with little or no vegetation |

Source: Adapted from Corine Land Cover.

Step 6: Localization and classification of the socio-environmental impacts and territorial conflicts

The objective is to identify the main impacts and territorial conflicts that affect the state of conservation of the core patches, the connectivity of the primary and secondary corridors and the quality of the landscapes that form the territorial matrix. This information will be obtained using the data from Step 5 and contrasted with fieldwork. The results are classified on four levels (Table 2) that condition the management guidelines of the GI Territorial Action Plan (Step 7):

Step 7: Creation of a GI Territorial Action Plan with management guidelines.

The guidelines will focus on planning multiple functions to improve sustainability and regional resilience. After carrying out the diagnosis in previous steps (4, 5, and 6), management guidelines will have to be drawn up considering the socio-environmental impacts and the territorial conflicts detected (Table 2) by the team of experts. Thinking in terms of regional resilience, actions should focus on the capacity of GI to solve important challenges such as climate change, food insecurity, and loss of biodiversity and limited natural resources. These actions should also identify the opportunities to develop a multi-scale GI that can contribute to the sustainable social and ecological health of the region. It will be necessary to incorporate a community-based strategy using participative techniques (round tables, focus groups, workshops, etc.) to plan the GI, looking for the representation of a wide range of agents (landowners, local communities, farmers´ associations, social networks, public administration, etc.). The participative techniques will allow, on the one hand, the planning of GI to be adapted to local priorities; and on the other hand, to increase the feeling of co-responsibility and acceptance from the local agents. These are essential aspects for ensuring the success of GI [44].

**Table 2.** Three types of territorial conflicts, according to status in order to maintain territorial connectivity and the delivery of ecosystem services of the matrix that makes up the GI.

| Conflict Classification | Description | Example |
|---|---|---|
| Moderate | These are dynamics that do not directly affect the core patches or ecological corridors but could undermine the state of the territorial matrix, and indirectly, the operation of the GI. They require sectoral regulations that are committed to territorial sustainability and adding value to the landscape and good agricultural practice. | –Increase in the surface area where changes are identified in types of production (horticultural crops to forage crops, loss of traditional mosaics). <br>–Possible embedding of agricultural intensification practices, which have a major impact on the landscape and the ecological operation of the region. <br>–Increase in processes of abandonment of agricultural activity with the encroachment of scrubland. <br>–Presence of reduced size of agricultural plots. <br>–Activation of erosion processes, with notable land loss. |
| Serious | These are dynamics that indirectly affect the core patches or ecological corridors of the GI. These dynamics need to be controlled with an up-to-date review of municipal planning, adjusted to existing values, such as permanent compliance with urban planning. | –Inadequate municipal urban planning for the formalization and correct operation of the GI. <br>–Modification of the layout of rural areas. <br>–Proliferation of dispersed housing and housing plots on rural land. <br>–Increase in dumping grounds and the accumulation of unregulated waste. <br>–Increase in hobby farming. |
| Very serious | These are dynamics that directly affect the core patches or ecological corridors of the GI. These dynamics need to be controlled with an up-to-date review of municipal planning, adjusted to existing values such as permanent compliance with urban planning. | –Existence of /increase in areas of mining-extraction. <br>–Permanent land sealing from new urban development. <br>–The existence /design and execution of lineal infrastructure with a major barrier effect. <br>–Alteration of the routes used for historical livestock routes and their improper use. <br>–Contamination of water courses and diseases suffered by the vegetation on their banks. |

Source: created by the authors.

### 2.3.2. Mapping the Functional Elements in the GI

In accordance with the postulates of landscape ecology, the GI must be made up of a series of elements that work together to favor the ecological and socio-ecological processes ranging from a local to a regional scale. The characteristics of the main elements of GI should be based on the functions that they have in the network. This characterization has been achieved by adapting the general proposal of the document, "Scientific and technical foundations for the Spanish strategy on green infrastructure and ecological connectivity and restoration" [52], and the proposals of other earlier studies [21,53]:

- Core patches: Are those spatial units in which the conservation of biodiversity is of prime importance for different species, even when dealing with areas which are not protected by law. They represent the anchorage to the network and can be of different shapes and sizes, as well as being both public and private areas: the Natural Protected Spaces, the Public Utility Forests and the Private Protected Forests have been used for the proposal for the formalization of the patches of GI as well as the spaces included in Natura 2000 network (ZEPAS, LIC/ZEC); the Community

Interest Habitats; and complemented with agroecosystems of great significance, or those that lack sectorial protection, such as the Important Bird Areas (IBAs).

- Ecological corridors: The objective of these green corridors is to maintain ecological and environmental connectivity by using physical links between the core areas. As with the core patches, they are of different shapes and sizes—even though they are generally linear in nature—and they can also be publicly or privately owned. In the region studied, these primary corridors are complemented with secondary ones, such as water courses and the network of historical livestock routes.

- Territorial matrix: In accordance with the landscape ecological planning approach, the territorial matrix constitutes the spatial-temporal basis resulting from the physical medium, the biological component, their functional relationships, and the transformations which human activity introduces into the system, which is expressed in the specific configurations of the landscape. Therefore, it is important to highlight the fact that the effectiveness of the ecological network is not found to such a great degree in the spatial entity of the core patches and the corridors, but in its ability to interconnect different noteworthy elements of the territorial matrix which are important for environmental matters, landscape, heritage and production in order to guarantee the corresponding ecological, economic and environmental processes and flows.

## 3. Results: Mapping the GI in the Comarca De Las Vegas

Step 1: Data compilation and creation of a database that is sufficiently detailed to be used at a supra-municipal scale and Step 2: Identification and characterization of landscape units which make up the territorial matrix of the selected study area.

The Comarca de las Vegas, is an agricultural regional unit with homogenous geographical and agricultural features. Following the methodology proposed, 16 landscapes were identified that belong to three large landscape types of the northwestern sector of the southern Iberian plateau, recognized in the Spanish Landscape Atlas [54]. These landscape units are (i) the *vegas regadas*, whose physiognomic and productive strength give the region its name; (ii) the *páramos* with their rain-fed herbaceous ligneous Mediterranean agriculture; (iii) the *cuestas* (Scarp slope), *escarpes* y *taludes* (Escarpments) and *vertientes*, with an agroforest mosaic of ligneous crops, shrubland (*Rhus coriaria, Genista scorpius, Asphodelus albus, Stipa tenacissima, Lepidium subulatum, Atriplex halimus, and Cistus clusii*), and hills of Holm oaks (*Quercus ilex*), Gall oaks (*Quercus faginea*), Kermes Oaks (Quercus coccifera), and spots of Aleppo pines (*Pinus halepensis*) which are as characteristic as they are ecologically valuable. The study of these landscapes distinguishes more than 100 sub-units in the region, a fact which expresses the internal diversity of this area (Figure 3).

Step 3: Selection and spatial delimitation of the functional elements for the formalization of GI

The study area possesses a group of high value natural spaces that are protected by different laws and specific protection instruments, which have been the basis of the proposal for the core patches of the GI. For the selection of the biggest core patches, different Protected Natural Spaces have been identified: the Regional Natural Park around the axis of the lower course of the rivers Manzanares and Jarama; the wildlife shelter on the San Juan Lake, in Chinchón; the Regajal del Mar de Ontígola Nature Reserve, in Aranjuez; different Public Utility Forests and the Private Protected Forests; and on a European level, the spaces which are part of the Natura 2000 network, which are complemented with Habitats of Community Interest. The protected spaces make up almost 50% of the surface area of the region. These previously mentioned core patches are complemented with agricultural landscapes of agroecological interest, such as the irrigated areas of the vegas, the Dehesa of Brea de Tajo, the Dehesa el Romeral in Aranjuez, the area used by large-scale agricultural and livestock farms of Valdealcalá in Ambite and all the farmland included in the IBA Baja Alcarria in Fuentidueña de Tajo, Estremera and Brea de Tajo.

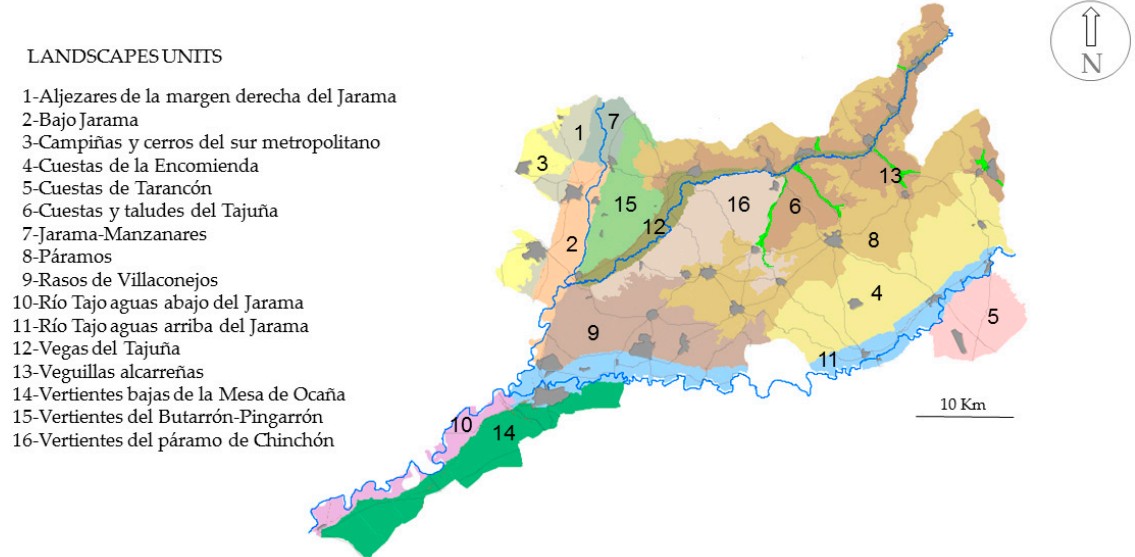

**Figure 3.** Landscape units of the Comarca de las Vegas. Source: based on criteria established by Mata, R. (Coord.) et al. [47].

The ecological corridors are delimited by the rivers and their associated tributaries which are extremely important ecological connectors because of their linearity, lengths, and routes. Historical livestock routes were also selected which used to play an important role in livestock transhumance, and currently guarantee mobility between agricultural plots, connecting, in the case study, important landscape units (*vegas* and *páramos, cuestas* and *vertientes*). These linear elements, together with the previously delimited ecological corridors [55], were mapped, developing a strategy to help the mobility of wildlife and guarantee the functionality and connectivity between the core patches (Figure 4). The landscape connectivity of this network can be supported by the presence of these dense hierarchical hydrographical networks (the River Tajo and its tributaries, Jarama and Tajuña) and an extensive system of historical livestock routes.

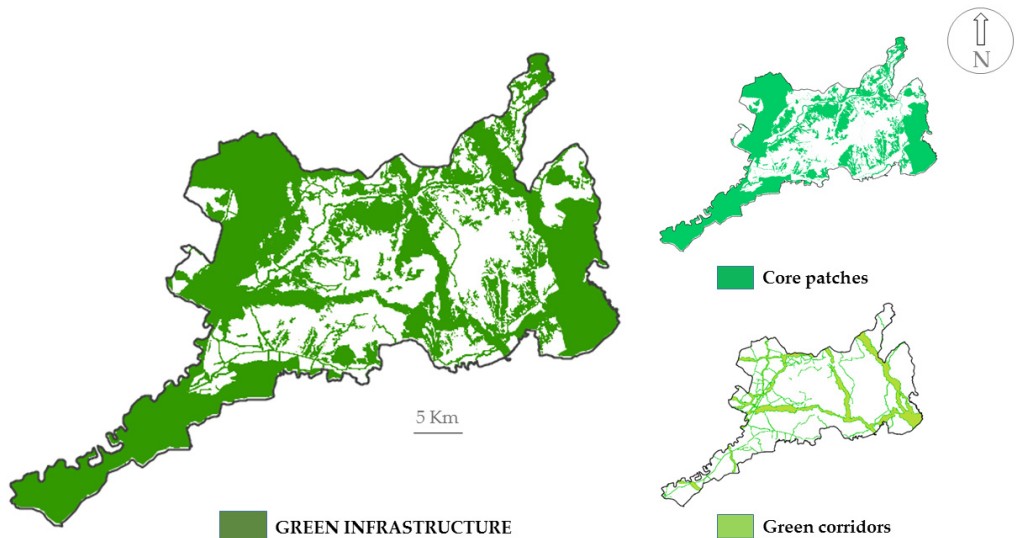

**Figure 4.** Green Infrastructure network map and its functional elements.

Step 4: Analysis of existing land use in the urban planning and creation of up-to-date mapping

The Autonomous Region of Madrid lacks a strategic plan that integrates environmental, economic, transport, and social dimensions as a whole, and that organizes urban growth and effectively addresses the conservation of open spaces. This means that las Vegas, as with other areas in Madrid, are affected by metropolitan urban sprawl which goes against the objective of territorial sustainability. Furthermore, the municipal urban planning of the 21 municipalities of the region is generally, obsolete. Only four of them have approved Urban Plans. The rest have Subsidiary Planning Rules (SPR), which form a lower ranking instrument with far fewer regulations. The existing urban instruments are very heterogeneous because their approval has taken place over a long period, from 1976 to 2016 (Table 3).

**Table 3.** Planning State, instrument type, and approval date.

| Municipalities | Instrument type | Approval date |
|---|---|---|
| Ambite | Subsidiary Planning Rules | 1995 |
| Aranjuez | Urban Plan | 1996 |
| Belmonte de Tajo | Subsidiary Planning Rules | 1999 |
| Brea de Tajo | Subsidiary Planning Rules | 1987 |
| Carabaña | Subsidiary Planning Rules | 1985 |
| Chinchón | Subsidiary Planning Rules | 1985 |
| Ciempozuelos | Urban Plan | 2008 |
| Colmenar de Oreja | Subsidiary Planning Rules | 1985 |
| Estremera | Urban Plan | 2012 |
| Fuentidueña de Tajo | Subsidiary Planning Rules | 1994 |
| Morata de Tajuña | Subsidiary Planning Rules | 1995 |
| Orusco de Tajuña | Subsidiary Planning Rules | 1997 |
| Perales de Tajuña | Subsidiary Planning Rules | 1978 |
| San Martín de la Vega | Subsidiary Planning Rules | 1997 |
| Tielmes | Subsidiary Planning Rules | 1976 |
| Titulcia | Subsidiary Planning Rules | 2000 |
| Valdaracete | Subsidiary Planning Rules | 1994 |
| Valdelaguna | Subsidiary Planning Rules | 1999 |
| Villaconejos | Subsidiary Planning Rules | 1985 |
| Villamanrrique de Tajo | Urban Plan | 2016 |
| Villarejo de Salvanés | Subsidiary Planning Rules | 2003 |

Source: created by authors using PLANEA.

The analysis and revision carried out using SIG, which has been combined with research work, show that the non-developable land with high protection accounts for 59% of the total surface area of the region (772 km$^2$) and the non-developable land with low protection represents 31.8% (417 km$^2$) (Figure 5). The rest (slightly less than 10%) is covered by, in general, urban land and developable land (with different degrees of consolidation and/or execution). Despite the high proportion of land which cannot be developed, the existing problem related to the planning law in force (Law 9/2001 from the Autonomous Region of Madrid) is that the land classified before 2001 as common non-developable land becomes considered as developable land which has not been divided into sectors, even though, in many cases, it has clear conservation merits. This is an extremely worrying reality, given that a notable amount of the land has agroecological significance and ecological value. These areas of land should be part of the future GI proposal.

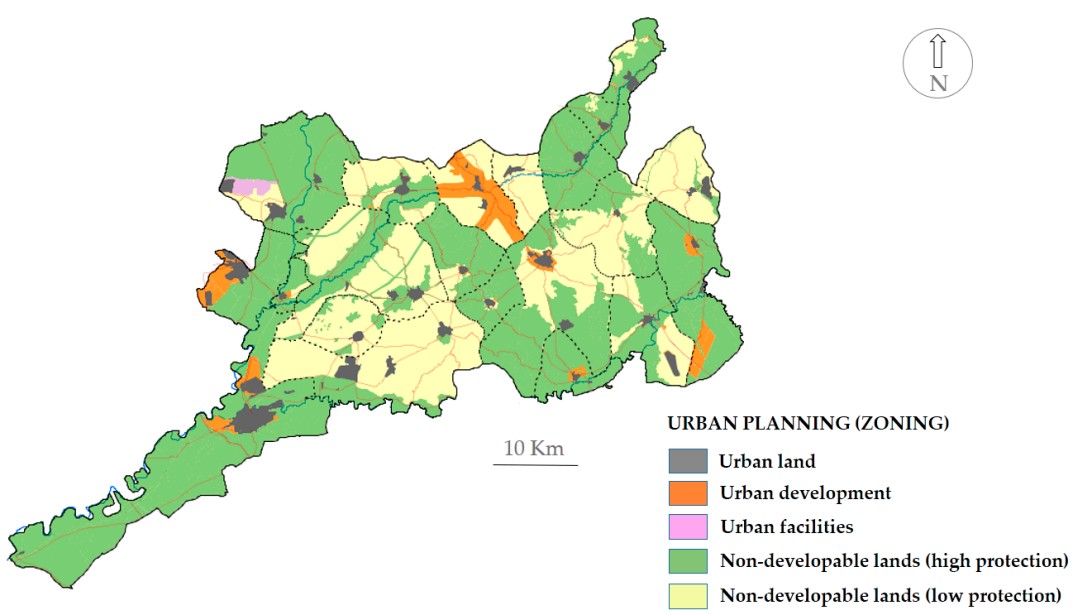

**Figure 5.** Land use in the urban planning of the 21 municipalities which make up the study area.

Step 5: Identification of land-use changes

In the period analyzed (1990–2018), in accordance with the information provided by CORINE Land Cover, a very significant increase in urban land can be appreciated in the region of las Vegas, a fact that must be considered when regarding the state of the territorial matrix and the implementation of GI (Figure 6). This process has been especially significant at the axis of the Jarama River (the municipalities of San Martín de la Vega Ciempozuelos and Aranjuez), with residential and commercial developments with high levels of land-use change. Moreover, this dynamic has meant the increase in certain areas of transport infrastructure. In the western part of the region, in specific municipalities whose functions are more rural than those previously mentioned, the presence of second homes and recreational areas have been noted, as well as some large scale facilities such as the Estremera prison (365,730 m$^2$), opened in 2008. Specifically, the artificial surface area occupied a total of 2.5% (33.59 Km$^2$) of the surface area of the region in 1990 and increased to cover 8.4% (110 km$^2$) of the total surface in 2018 (Table 4).

**Table 4.** Changes between 1990 and 2018.

| Land-Use/Cover | 1990 | % | 2018 | % |
|:---:|:---:|:---:|:---:|:---:|
| Artificial surface area | 33 km$^2$ | 2.5 | 110 km$^2$ | 8.4 |
| Agricultural areas | 838 km$^2$ | 64.1 | 748 km$^2$ | 57.2 |
| Forest and semi-natural areas | 437 km$^2$ | 33.4 | 450 km$^2$ | 34.4 |
| **Total** | **1308 km$^2$** | **100** | **1308 Km$^2$** | **100** |

Source: created by authors using Corine Land Cover.

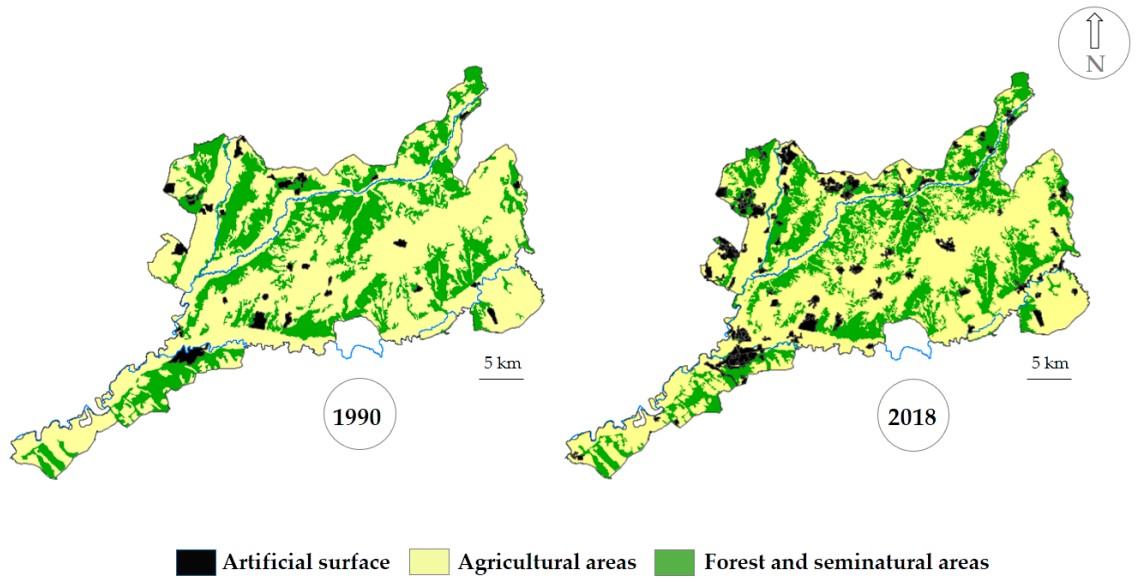

**Figure 6.** Land-use changes between 1990 and 2018 in the Comarca de las Vegas.

On the other hand, a slight reduction in agricultural terrain has been noted, which can be explained by the selective abandonment of certain traditional farms, those with little profitability, struggling to adapt to market demands and the weak agricultural policies of the Autonomous Region of Madrid. The abandonment of the rain-fed land has been accompanied by a process of scrub encroachment identified in the fieldwork and in the analysis of land cover from the information provided by Information System of Land Occupation of Spain (SIOSE) (scale 1:25.000). Another important dynamic is the change in the type of crops on the alluvial plains, with a clear reduction in the production of horticultural crops and an increase in forage crops, implemented in large-scale farms, with irrigation and a high level of mechanization (Figure 7). Finally, forest coverage, when considered as a whole, has slightly increased, even though some natural coverage on riverbanks has been lost.

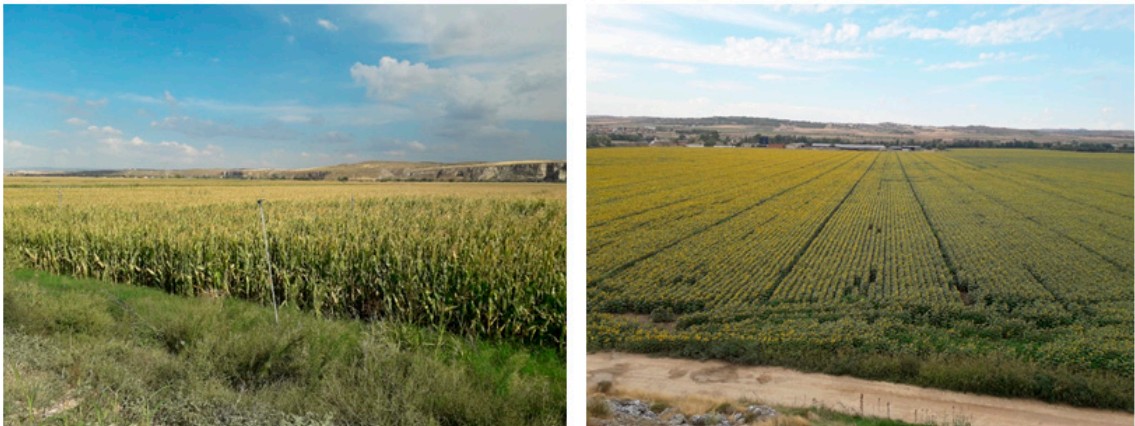

**Figure 7.** Forage crops (maize on the left, sunflowers on the right), in large-scale farms with irrigation on the alluvial plain of the River Tajo.

Step 6: Localization and classification of the socio-environmental impacts and territorial conflicts that affect the state of conservation of the core patches and the connectivity of the primary and secondary corridors.

Landscape connectivity in the region of las Vegas is altered by changes in land cover and land use, including the fragmentation of habitats, an increase in patch isolation and the loss of ecological connectivity. Serious fragmentation processes can be observed which have been caused by urbanization processes and especially by the densification of transport and communication infrastructure (high-capacity roads, conventional roads, and railways lines) (Figure 8). One of the main problems detected during the research is fragmentation. This doubly conditions the normal development of agricultural activity and the capacity to supply ecosystem services (agro-biodiversity, food production, soil infiltration, visual quality, etc.) produced mainly by the modification of the layout of rural plots and the deterioration of specific agricultural paths and livestock routes. It is well known that the fragmentation caused by the road network increases labor costs as the time needed to work on different, distant small-scale plots increases, favoring the abandonment of agricultural activity [56]. Other realities which notably condition the conservation of the agrarian matrix and the quality of the ecosystem services is open pit mining and, to a lesser extent, the dumps and areas of illegal dumping (Figure 9). All of these situations require urgent measures regarding protection, mitigation, and restoration (Table 5), which will have to be established in the action plan that is currently being worked on.

**Table 5.** Classification of the main territorial conflicts, management measures and priority areas.

| Impact | Conflict Classification | Measures | Priority |
|---|---|---|---|
| Mining areas and/or dumps in the GI | Very serious | Ecological and landscape restoration. | Areas in the Southeastern Regional Natural Park (the Jarama River axis). |
| Infrastructure that crosses ecological corridors in the GI, creating a barrier effect | | Improvement in permeability (wildlife crossings). | Areas affected by the barrier effect produced by high capacity routes (highways and railways). |
| Areas with inadequate municipal planning for GI | Serious | Revision of municipal planning in accordance with measures established in a high-level territorial plan. Inclusion of land in the protected non-developable land category. In some cases, proposal for the declaration of Protected Natural Spaces to be made to the sectoral administration (*) | (*) Dehesa de Brea de Tajo and terrain included in the IBA. |
| Possible impacts on areas of high agricultural and landscape interest | Moderate | Creation of agrarian parks. Activation of territorial land stewardship strategies or contracts. | Areas located in *páramos* and *rasos* and in the *vega media* of the Tajuña. |

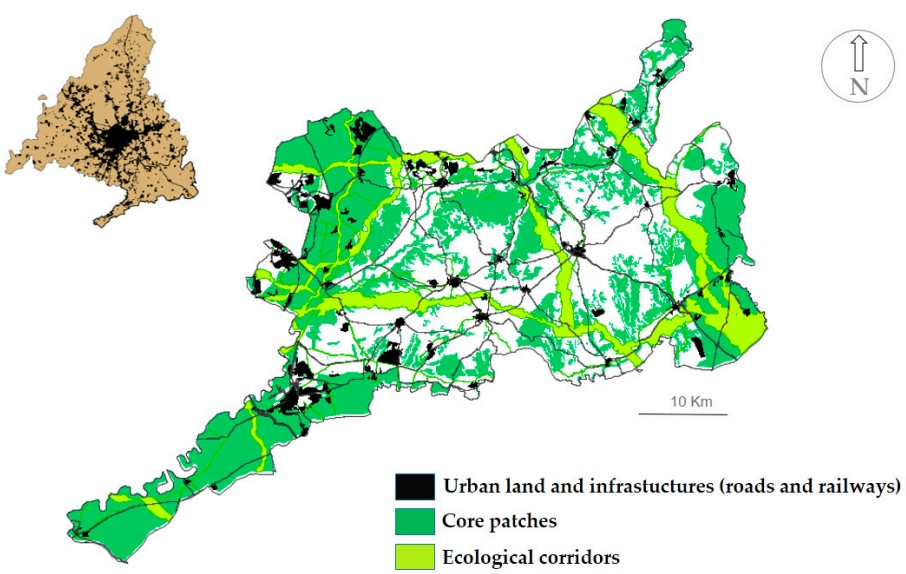

**Figure 8.** Fragmentation of habitats and loss of ecological connectivity caused by urban development and infrastructure.

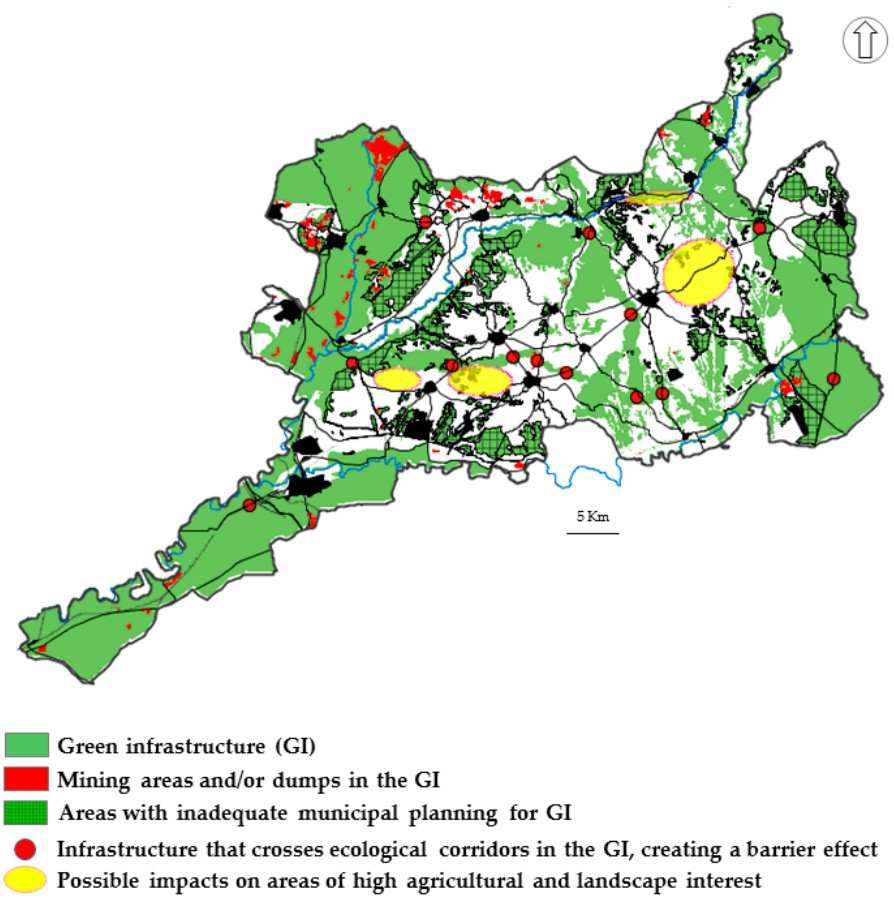

**Figure 9.** Territorial conflicts that affect the territorial connectivity and the delivery of ecosystem services of the matrix that makes up the GI.

Step 7: Creation of a Territorial Action Plan of the GI with management guidelines.

The obsolescence of municipal planning, the absence of a territorial planning instrument at a supra-municipal scale and the numerous impacts detected mean that the approval of a Territorial Action Plan must be a priority for the Comarca de Las Vegas. The development of this strategic plan must be based on territorial political legislation concerning territorial and urban planning and the legislation on protected natural spaces of the Autonomous Region of Madrid. The plan must incorporate the priority management guidelines regarding open spaces focusing on the following elements:

1.  Establishment of explicit protection for the GI proposed, with its different functional elements (core patches and ecological corridors), creating a specific planning category (Figure 4).
2.  The ecological corridors, regardless of their urban planning classification, must be defined and organized according to territorial and urban planning guidelines to guarantee functional and ecological connectivity.
3.  Incorporation of stricter regulations on agricultural land usage. Zoning designation should strictly limit any non-farming usage that competes with food production. Land-use planning should improve and protect valuable agricultural landscapes that form the matrix, without which the GI would not be able to perform its functions and provide ecosystem services (food provision, regulation, and cultural services).
4.  An increase in the level of protection of specific spaces, insisting that the environmental administrative bodies integrate these functional spaces into the regional or European network (Red Natura 2000) of protected natural spaces (for example, the Dehesa de Brea de Tajo; the Dehesa del Romeral in Aranjuez; Monte Valdealcalá, in Ambite and the IBA "Baja Alcarria" in Fuentidueña de Tajo, Estremera and Brea de Tajo).
5.  Creation of a series of specific measures to improve the GI functions aimed at the recovery of degraded land caused by mining activities, the regeneration of certain extremely valuable ecosystems found in core patches and limiting the expansion of forest land over farmland. Strategically siting new transportation facilities (road, railway, and metro networks) to prevent landscape fragmentation.
6.  Incorporation of a series of initiatives to activate the territory in terms of its landscape values and production capacity. By promoting policies for incentivizing local horticulture production and the stimulation of new generation farmers to contribute to the improvement of food self-sufficiency and food security and boosting endogenous development. Therefore, it seems important to create some small-scale agrarian parks and to encourage the adoption of different land stewardship contracts. Considering the agroecological characteristics and the agrarian landscape diversity of the Comarca de Las Vegas, the future agrarian parks could be situated in the *vega media* of the Tajuña, in the *rasos* of Villaconejos and Colmenar de Oreja and in the *páramos* of Valdaracete-Villarejo de Salvanés.

## 4. Discussion

We present a multi-scalar and landscape ecology approach for planning multifunctional GI at a regional scale. The main, and most innovative contribution is that it gives a strategic role to peri-urban agriculture and traditional agricultural landscapes. These areas do not usually have proactive management despite this being essential for maintaining the quality of open spaces and the provision of ecosystem services. In contrast with the more frequent design proposals for the GI network, which bestow, almost exclusively, importance on the core patches and the ecological corridors, our methodology gives a proactive approach to the agrarian-based socio-ecological matrix. By ensuring farmland preservation and promoting sustainable farming practices, the capacity of agro-ecosystems to supply food production and raw materials can be achieved. This will also ensure the maintenance of soil fertility, agro-biodiversity, pollination, and cultural services.

This methodology is also useful for identifying hotspots, information which enables planners to mitigate landscape problems such as fragmentation. The methodology proposed adapts the scale and focus of landscape ecology into the design and the management of GI, which can improve the functionality of the green network and incorporate a wider range of environmental and socio-economic benefits. The methodology also delivers the appropriate development of multifunctional GI resources. It also has the advantage of being inherently planned using web-based information that is available on, or downloadable from, the mapping viewers provided by the public administration bodies (roads, protected natural areas, rivers, etc.) and by using readily available datasets such as CORINE land cover. Another advantage is that it can be used with free GIS software (GVSIG and QGIS). Therefore, this methodology is easily applicable for the identification and evaluation of the elements being integrated into GI by planners, territorial managers, and community organizers, or by experts from public entities whose resources and funding tend to be limited.

Nevertheless, this framework has some limitations. One of these, is that the identification of the impacts and changes in land use/cover is based on GIS information. It is notable Corine Land Cover captures some changes relatively well (e.g., changes in urban land extension), but estimates for some key land-use change processes, for example agricultural abandonment or deforestation, do not receive the in-depth treatment they need [51] (p. 2). Furthermore, the rates and patterns of changes in the management intensity of agriculture and forestry are not captured by CORINE Land Cover [50]. These problems can be solved by contrasting the results with other more precise land-cover databases, for example, in Spain, by using the SIOSE and largely through fieldwork. Another limitation could be that the methodology largely depends on web data availability, at the proper scale of analysis. Finally, we can indicate that the GI alone does not have the capacity to overcome all territorial challenges, which are mainly due to urbanization processes and urban sprawl. The effectiveness of GI to supply the provision, regulation, and cultural ecosystem services, will be extremely limited unless there is major political and financial support.

The in-depth analysis, using the methodology followed during the case study, reveals that some areas in las Vegas have a greater need for the implementation of GI than other parts of the region. This is due to a weak planning strategy at a supra-municipal scale which has neither reduced the urban pressures of rural areas close to the capital nor efficiently protected the important landscape values. This situation is worsened by the fact that the municipalities which make up the study area, apart from a limited number of exceptions, lack urban plans adapted to their current requirements, as the existing ones were drafted in the 90s or earlier (7 of the 21 municipalities have urban plans from before 1990) (Table 3). Nevertheless, all the changes in land use in the region between 1990 and 2018 are serious, as detected in the analyses carried out. Urban development has appeared as it commonly does, with an increase in the size and density of the road network. This dynamic is inseparable from the metropolitan phenomenon and the extension southwards of the urban print of the city of Madrid, which generates extreme pressure on the peri-urban spaces. In the same period, the agricultural surface area has experienced a slight decline, which can be explained by the selective abandonment of certain traditional farms, which generally occupied marginal terrains on the edges (in *páramos* and *vertientes*). The abandonment of these rain-fed areas means an increase in the encroachment of shrubland, identified in the fieldwork and mapping work. Even though the cultivated zones have not experienced significant changes in their extension in the last two decades, they have experienced a change in the type of production in favor of extensive and mechanized crops, with the incursion of forage and industrialized crops in las Vegas, especially in those of the rivers Jarama and Tajo, with a major presence of maize, and on a smaller scale, sunflowers, on land which, in part was historically for horticultural use.

In the current context, the limited availability of fertile land in the metropolitan areas and the problems related to the complex rural–urban dynamics mean that it is necessary to employ a multifaceted approach of various policy measures [57]. In this sense GI is able to integrate a wide range of issues and strategies instead of conventional forms of urban planning. In order to achieve

this, GI can be complemented with an agreed Territorial Action Plan. Moreover, many of the problems and impacts on peri-urban farmland can be addressed, as indicated by Meerow and Newell [58], by combining different planning criteria and ranking them according to the priorities of local stakeholders, and combining these criteria with a GI spatial model which can enable planners to identify hotspots where GI has the greatest potential to foster social and ecological resilience. The scale of implementation should be significant enough to introduce rationality and balance between different forms of land use and offer innovative solutions for land-use planning, which recognizes the value of the traditional agricultural landscapes.

Therefore, to improve the success of the implementation of GI, it must be supported by territorial planning measures concerning open spaces, public support for the planning, and zoning of agricultural areas, as well as being supported by governance at different scales of action. The productive potential of agricultural land must be protected to improve land-use planning to support sustainable food systems, [59]. Regarding effectiveness of GI to fulfil this objective, GI can be supported by territorial tools which have similar objectives, such as the agrarian parks and land stewardship contracts. The agrarian parks included in GI could be beneficial given that they are instruments focused on protecting fertile land, improving economic activity linked to local agriculture and strengthening the multifunctional character of the agricultural spaces. Moreover, land stewardship contracts can help improve the commitment of owners and farmers to the maintenance of the landscape and the conservation of the habitats.

## 5. Conclusions

We have explored the different functions and the scales involved in implementing GI projects in recent literature. Current research indicates that GI is an effective tool for improving the traditional approach to the protection of biodiversity and landscape connectivity. Moreover, this research highlights remarks that GI improves ecosystem management when it adopts a socio-ecological approach. In relation to developing sustainable planning strategies, firstly, our findings show that when designing GI at a supra-municipal scale, it is possible to integrate landscape, environmental, social, and economic services, as a whole. Secondly, it is possible to improve the territorial connectivity between the different landscapes that make up the territorial matrix when taking advantage of the key components of open spaces. Thirdly, at the supra-municipal scale GI is suitable for the management of green spaces as it improves recreation, public access and enjoyment, and conservation. These are aspects of great importance, especially at a metropolitan scale, where there is a convergence of different pressures caused by urban sprawl and the increase in the densification of transport infrastructure.

Nevertheless, we have observed that there is a gap in recent literature regarding the treatment of agriculture spaces and the role given to them in GI, which must be filled. This issue stands out as there is a wealth of evidence on the negative impacts which are caused by the absence of, or shortage of protection and management strategies of farm land and especially those which are located on the urban fringe.

Protected agricultural land without spatial barriers is needed to assure food supply and food security. To respond to this challenge, a methodology based on the landscape ecological planning approach has been proposed giving greater protagonism to the agrarian-based socio-ecological matrix. By improving the multi-functional treatment of the different agricultural landscapes, a wide variety of ecosystem services and functions can be delivered. This is key for ensuring the effectiveness of land-use planning that promotes sustainability and the resilience of metropolitan areas, especially in the face of rapid land-use changes. As well as having public support at different scales and stronger strategic planning for protecting and supporting traditional agrarian landscapes. This will contribute to slowing down or reversing the process of agricultural land loss. Ultimately, it is about the reassessment of multi-functional agricultural landscapes in their different contexts, and the reconstruction of the connection between agriculture and the urban areas by creating specific economic incentives (short-food supply chains, farmers' markets etc.). With the focus on increasing

the social and ecological sustainability of the green network as a whole, landscape connectivity and food security can be improved.

**Author Contributions:** This paper is the result of collaborative teamwork. Conceptualization, C.Y.O. and R.M.O.; methodology, C.Y.O., R.M.O., and D.F.J.; writing—original draft preparation and supervision, C.Y.O.; writing—review and editing, C.Y.O., R.M.O., and D.F.J. All authors have read and agreed to the published version of the manuscript.

**Funding:** This research has received funding from the research project "Paisaje y Huerta de Madrid" (PDRR-I8 agreement Autonomous University-IMIDRA) co-financed by the European Union through the European Agricultural Fund for Rural Development, Spanish Ministry of Agriculture, Fisheries, Foodstuffs and the Environment and the Community of Madrid-IMIDRA Rural Development Program 2014–2020, and the ongoing research project SAMUTER from the European Agricultural Fund for Rural Development and the Spanish Ministry of Agriculture, Fisheries, Foodstuffs and the Environment, in the 2018 call, submeasure 16.1 within the framework of National Rural Development Program 2014–2020.

**Conflicts of Interest:** The authors declare no conflict of interests. The design of the study; the collection, analyses, or interpretation of data and writing of the manuscript were the responsibility of the authors.

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
