# Peer review of "Green Infrastructure Planning in Metropolitan Regions to Improve the Connectivity of Agricultural Landscapes and Food Security"

_land, doi:10.3390/land9110414_

Round 1
Reviewer 1 Report
Thank you very much for this very interesting read worth for publication after some revisions. It is more than just minor editorial revisions but not a very deep revision. Some more additional reflections in the discussion and conclusions section would improve the paper significantly. Besides spelling and some small additional explanations (see my descriptions by lines below), my main points are:
Methodology:
Literature review – add some lines how you avoid bias the way how you conducted and selected the literature.
Discussion:
Try to add some lines on your methodology to a broader context than Spain and applications in other regions. You already have plenty of things that could work elsewhere, e.g. CORINE with its advantages and disadvantages, data banks on land uses available (or not). Instead of reading them between the lines, put them more prominently.
Although it gets a bit political, I personally miss some lines on how GI planning can be strengthened and how to get GI protected and certain agricultural practices becoming more legally binding in the discussion section. Perhaps, look/compare to other countries/regions that have stronger, more powerful, more binding planning frameworks. But to which extent such tools can work? Can such tools, or a protection status exclude the cultivation of certain crops or request certain agricultural practices to secure provisioning of certain ecosystem services?
Conclusions:
Related to the point before, for the conclusion section, I would also add some lines, on what would you suggest to support a better GI planning and to better protect GI and agriculture from the pressure of urbanization? Would it be planning tools, a stronger regional planning, more legally binding classifications?
Checking some of the references, it seems to me that at least some of them obviously got mixed or shifted when preparing or editing the references and respective numbers. Although this number format is quite error-prone, many journals want to have it this way. So please go through the references and check them thoroughly to have your references correct.
Here are some small points by lines:
Introduction:
Lines 32-35: I´d suggest to slightly re-write the section as it does not fully reflect the situation in different places on the globe. Open space can be agricultural land, unfertile land or managed /unmanaged forests depending on the geographical region you are in. Also there is a discussion about the different concepts of Ecosystem Services and Functions and how to separate/distinguish the concepts – to avoid opening up this discussion or expectations in this direction, I´d suggest to re-write the section as following:
… multi-functional focus [1-6]. Open space is a term which can be used in a broader sense to describe areas with a low level of human intervention. These areas of land are increasingly becoming key pieces of metropolitan planning [2]. Regarding open space planning, the provision of ecosystem services stands out in developing GI [7-12].
Line 59: “produce” check language – products (?), I am not a native English speaker – so check if your wording is might be better suiting than my suggestion.
Line 84: “non developable land” – I would write open space /of land set aside from urban development. This might perhaps be better as your current wording might be a bit misleading, as you write about managing this land later with is, strictly speaking, this is some kind of development in terms of agricultural uses.
Line 108: Scalar – scale(?). Again, I am not an English speaker and not sure. If you opt to change the term, do this stringent throughout the text (e.g. line 157)
Line 114: y - and
Line 124: Literature citation is not correct.
Line 116-126: Needs to be explained a bit more in a few words or lines (not long), especially to the point how to ensure avoidance of bias this way (see my introductory remarks)
Lines 141-144: A bit more description of the urbanization pressure to this area would be good to understand the situation. It does not need to be long, only briefly, just a few lines.
Line 240: Participative techniques – do you mean approaches (?), otherwise, it would be interesting to briefly mention some of them e.g. in a bracket (e.g. …)
Line 298-299: Add Latin names for tree species.
Line 306-307: “ needs to be put correctly
Line 309-310: new features – although somewhat obvious, please briefly list some of them with “such as…” or in brackets (e.g. …)
Line 349: Are you sure the cited literature is correct or something got mixed or shifted? The literature source you refer to is English, the landscape units are in Spanish
Line 435: delete one of the points
Line 455: Graph: Legend: Railways (letter missing)
Line 479: delete blank(s).
Author Response
We thank the reviewers for their constructive comments, which have been helpful in improving the quality of
The manuscript. We hope we have adequately addressed all the points raised by the reviewers. Please find in the attachment
our detailed responses to reviewers' comments.

Reviewer 2 Report
Even if the paper addresses an interesting topic, there are several issues that need to be strengthened/reviewed on the paper, namely:
Literature review misses several seminal works and important advances crossing GI management. Read for example (Ferreira or Panagopoulos, regarding this subject).
Material and methods are hard to understand, considering the fuzziness of descriptions associated to each research step. I would recommend the introduction of a phased methodological diagram, and the introduction of a specific materials and methods chapter.
Conclusions need to be more scientific. As they are they highlight the limitations of the research... Further information is needed. Even empirical conclusions, need to be supported on specific data.
Author Response
We thank you for your constructive comments, which have been helpful in improving the quality of the manuscript. We hope we have adequately addressed all the points raised by the reviewers. Please find below our detailed responses to reviewers' comments.

Reviewer 3 Report
The manuscript proposes a methodology to contribute to integrating GI planning in metropolitan regions. In the light of global change, this important topic needs to be addressed to assure the provision of crucial ecosystem services to the residents and to foster a sustainable development.
Major comments:
While I think that the mapping and spatial analysis provides valuable information, I am not convinced that the review is relevant for the current research. Therefore, I suggest removing this study aim and section 2.1 respectively. Most of the arguments are already presented in the introduction and I see no further benefit of describing the details of the literature review. To improve understanding and reading flow, I also suggest starting the section 2 with the description of the study area (now section 2.4), followed by section 2.2 and then section 2.3.
The proposed framework looks at the first glance convincing, but at the end of the results section, I wondered why you carried out all these steps. The last map (and central result (?), fig 9) is just a combination of GI elements and urban land and infrastructure. It seems that the final (and crucial) step is missing. Currently, the analysis remains on a very basic and descriptive level without an in-depth analysis and without combining the intermediate results. My central question is therefore, how do you finally combine all information of the seven steps? Where is step 7? The results as they are presented now are providing not much new information and only a combination of all influencing factors could in my opinion provide deeper information on where and how to develop mitigation measures and to develop GI.
Another concern I have is that it remains unclear how GI contributes to the provision of ecosystem services as mentioned in the introduction (and discussion). It is clear that GI is important for biodiversity conservation, but how is GI related to ecosystem services? How are ecosystem services influenced by GI? And where is this connection included in the proposed framework (and analysis)? This needs to be clarified or less emphasized.
The discussion section needs to be substantially revised. Please discuss your own results (without repeating them) in relation to other studies and indicate also the limitations of your study. It is also not clear how L470-492 are connected to your findings. If you could address my previous concerns, I think this would provide a valuable discussion basis, where you could also think of possible planning tools and management solutions.
In general, the manuscript reads well, but it is also very long and contains wordy parts. Please adopt a more concise writing style and remove unnecessary words or sentences.
Minor comments:
Section 2.3.1: It would be helpful for the reader to have a summarizing flow-chart of the seven steps at the beginning of this section (calling them ‘step 1’, ‘step 2’… as used in the results). Currently, this section describes mainly the aim of the single steps, but it remains unclear how you performed each step. Please add further methodological details. Moreover, I suggest using a short heading for each step, then describing shortly the aim and rational and including full methodological details, e.g. which CORINE classes, how you derived visual classes, how you carried out the fieldwork…
L175-179: It is not clear whether the first step aims at delimiting the study area (which in my opinion is not necessary) or at identifying landscape units. Please revise clarify and restructure these two sentences.
L230: Remove “This table shows the” from the caption.
Section 2.3.2: Did you delimitate these functional elements by yourself or were these spatial datasets already available? Please clarify and add references or a description of the analysis.
Figure 4: This figure is almost identical to Figure 1. I suggest or using a table, integrating it into Figure 5 or moving it to the supplementary material.
Figure 7: I suggest using a darker green for the forest to improve readability. Moreover, it would be nice to have a map of the changes to directly see the developments.
L406: Please use the term ‘land cover’ or ‘land use/cover’ throughout the manuscript, as CORINE presents land cover and does not provide information on land use.
L417-418: I don’t understand whether urban areas covered 33 or 110 km². Please revise. I would also be more useful to indicate %-changes between 1990 and 2000 for the major land cover changes.
Author Response
We thank you for your constructive comments, which have been helpful in improving the quality of The manuscript. We hope we have adequately addressed all the points raised by the reviewers. Please find below our detailed responses to reviewers' comments.

Round 2
Reviewer 3 Report
The revisions have improved the manuscript, but several issues were not sufficiently addressed.
One major issue is related to the methodology. I still have difficulties to understand how the single steps are connected to each other and how the information is finally combined in step 7. For example, how do you include the analyzed land-use changes, how do you deal with fragmentation? The various steps produce different maps, but they are never overlaid. In the results section, the last step is rather a summary of recommendations, but it does not explain how the generated results are used. Please make another effort to clearly describe this.
In relation to the previous comment, in section 2.3, it would be therefore helpful adding a summarizing flow-chart of the seven steps, as already indicated in the first review round.
The steps should also be indicated in the results section to relate the results to the methodological steps more easily.
The other issue is still the literature review. While I think the introduction has more substance after integrating some results from the literature review, I really don’t see the benefit of including the details of how you carried out the review in the manuscript and your response did not convince me. In my opinion, it would be sufficient to indicate in section 2.3 that the methodology you propose here is based on a literature review, but without overloading the manuscript with irrelevant details. I would rather suggest to place the first part of the discussion (L500-528) at the beginning of section 2.3., where it would be a good justification of the proposed methodology (and I understand that you would like to include it in the paper, while I don’t think that a result should be part of the discussion.
Minor comment:
Figure 6: As mentioned in the first review round, a map indicating the land-use changes would be more helpful to directly see the developments.
Author Response
Again we want to thank you for your comments, which have undoubtedly improved the quality of our manuscript. We hope we have resolved all comments.
Please see the attachment.
